# Neural Controlled Differential Equations for Irregular Time Series

**Patrick Kidger**  **James Morrill**  **James Foster**  **Terry Lyons**

Mathematical Institute, University of Oxford
The Alan Turing Institute, British Library
{kidger, morrill, foster, tlyons}@maths.ox.ac.uk

## Abstract

Neural ordinary differential equations are an attractive option for modelling temporal dynamics. However, a fundamental issue is that the solution to an ordinary differential equation is determined by its initial condition, and there is no mechanism for adjusting the trajectory based on subsequent observations. Here, we demonstrate how this may be resolved through the well-understood mathematics of *controlled differential equations*. The resulting *neural controlled differential equation* model is directly applicable to the general setting of partially-observed irregularly-sampled multivariate time series, and (unlike previous work on this problem) it may utilise memory-efficient adjoint-based backpropagation even across observations. We demonstrate that our model achieves state-of-the-art performance against similar (ODE or RNN based) models in empirical studies on a range of datasets. Finally we provide theoretical results demonstrating universal approximation, and that our model subsumes alternative ODE models.

## 1 Introduction

Recurrent neural networks (RNN) are a popular choice of model for sequential data, such as a time series. The data itself is often assumed to be a sequence of observations from an underlying process, and the RNN may be interpreted as a discrete approximation to some function of this process. Indeed the connection between RNNs and dynamical systems is well-known [1, 2, 3, 4].

However this discretisation typically breaks down if the data is irregularly sampled or partially observed, and the issue is often papered over by binning or imputing data [5].

A more elegant approach is to appreciate that because the underlying process develops in continuous time, so should our models. For example [6, 7, 8, 9] incorporate exponential decay between observations, [10, 11] hybridise a Gaussian process with traditional neural network models, [12] approximate the underlying continuous-time process, and [13, 14] adapt recurrent neural networks by allowing some hidden state to evolve as an ODE. It is this last one that is of most interest to us here.

### 1.1 Neural ordinary differential equations

Neural ordinary differential equations (Neural ODEs) [3, 15], seek to approximate a map $x \mapsto y$ by learning a function $f_\theta$ and linear maps $\ell_\theta^1$, $\ell_\theta^2$ such that

$$y \approx \ell_\theta^1(z_T), \quad \text{where} \quad z_t = z_0 + \int_0^t f_\theta(z_s)\mathrm{d}s \quad \text{and} \quad z_0 = \ell_\theta^2(x). \tag{1}$$

Note that $f_\theta$ does not depend explicitly on $s$; if desired this can be included as an extra dimension in $z_s$ [15, Appendix B.2].

Neural ODEs are an elegant concept. They provide an interface between machine learning and the other dominant modelling paradigm that is differential equations. Doing so allows for the well-understood tools of that field to be applied. Neural ODEs also interact beautifully with the manifold hypothesis, as they describe a flow along which to evolve the data manifold.

This description has not yet involved sequential data such as time series. The $t$ dimension in equation (1) was introduced and then integrated over, and is just an internal detail of the model.

However the presence of this extra (artificial) dimension motivates the question of whether this model can be extended to sequential data such as time series. Given some ordered data $(x_0, \ldots, x_n)$, the goal is to extend the $z_0 = \ell_\theta^2(x)$ condition of equation (1) to a condition resembling "$z_0 = \ell(x_0), \ldots, z_n = \ell(x_n)$", to align the introduced $t$ dimension with the natural ordering of the data.

The key difficulty is that equation (1) defines an ordinary differential equation; once $\theta$ has been learnt, then the solution of equation (1) is determined by the initial condition at $z_0$, and there is no direct mechanism for incorporating data that arrives later [4].

However, it turns out that the resolution of this issue – how to incorporate incoming information – is already a well-studied problem in mathematics, in the field of rough analysis, which is concerned with the study of *controlled differential equations*.[1] See for example [16, 17, 18, 19]. An excellent introduction is [20]. A comprehensive textbook is [21].

We will not assume familiarity with either controlled differential equations or rough analysis. The only concept we will rely on that may be unfamiliar is that of a Riemann–Stieltjes integral.

## 1.2 Contributions

We demonstrate how controlled differential equations may extend the Neural ODE model, which we refer to as the *neural controlled differential equation* (Neural CDE) model. Just as Neural ODEs are the continuous analogue of a ResNet, the Neural CDE is the continuous analogue of an RNN.

The Neural CDE model has three key features. One, it is capable of processing incoming data, which may be both irregularly sampled and partially observed. Two (and unlike previous work on this problem) the model may be trained with memory-efficient adjoint-based backpropagation even across observations. Three, it demonstrates state-of-the-art performance against similar (ODE or RNN based) models, which we show in empirical studies on the CharacterTrajectories, PhysioNet sepsis prediction, and Speech Commands datasets.

We provide additional theoretical results showing that our model is a universal approximator, and that it subsumes apparently-similar ODE models in which the vector field depends directly upon continuous data.

Our code is available at `https://github.com/patrick-kidger/NeuralCDE`. We have also released a library `torchcde`, at `https://github.com/patrick-kidger/torchcde`

## 2 Background

Let $\tau, T \in \mathbb{R}$ with $\tau < T$, and let $v, w \in \mathbb{N}$. Let $X: [\tau, T] \to \mathbb{R}^v$ be a continuous function of bounded variation; for example this is implied by $X$ being Lipschitz. Let $\zeta \in \mathbb{R}^w$. Let $f: \mathbb{R}^w \to \mathbb{R}^{w \times v}$ be continuous.

Then we may define a continuous path $z: [\tau, T] \to \mathbb{R}^w$ by $z_\tau = \zeta$ and

$$z_t = z_\tau + \int_\tau^t f(z_s) \mathrm{d}X_s \quad \text{for } t \in (\tau, T], \tag{2}$$

where the integral is a Riemann–Stieltjes integral. As $f(z_s) \in \mathbb{R}^{w \times v}$ and $X_s \in \mathbb{R}^v$, the notation "$f(z_s)\mathrm{d}X_s$" refers to matrix-vector multiplication. The subscript notation refers to function evaluation, for example as is common in stochastic calculus.

Equation (2) exhibits global existence and uniqueness subject to global Lipschitz conditions on $f$; see [20, Theorem 1.3]. We say that equation (2) is a controlled differential equation (CDE) which is controlled or driven by $X$.

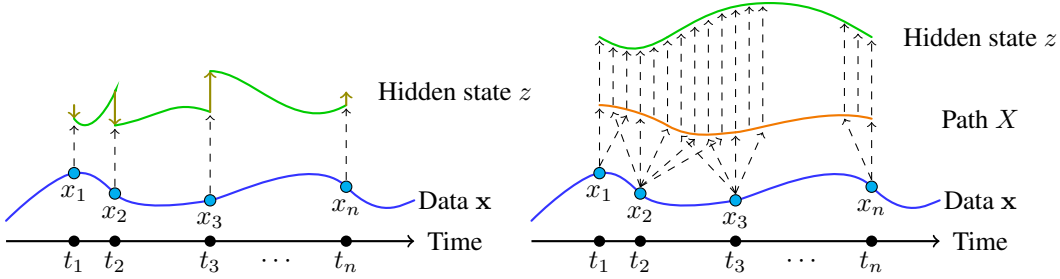

Figure 1: Some data process is observed at times $t_1, \ldots, t_n$ to give observations $x_1, \ldots, x_n$. It is otherwise unobserved. **Left:** Previous work has typically modified hidden state at each observation, and perhaps continuously evolved the hidden state between observations. **Right:** In contrast, the hidden state of the Neural CDE model has continuous dependence on the observed data.

## 3 Method

Suppose for simplicity that we have a fully-observed but potentially irregularly sampled time series $\mathbf{x} = ((t_0, x_0), (t_1, x_1), \ldots, (t_n, x_n))$, with each $t_i \in \mathbb{R}$ the timestamp of the observation $x_i \in \mathbb{R}^v$, and $t_0 < \cdots < t_n$. (We will consider partially-observed data later.)

Let $X \colon [t_0, t_n] \to \mathbb{R}^{v+1}$ be the natural cubic spline with knots at $t_0, \ldots, t_n$ such that $X_{t_i} = (x_i, t_i)$. As $\mathbf{x}$ is often assumed to be a discretisation of an underlying process, observed only through $\mathbf{x}$, then $X$ is an approximation to this underlying process. Natural cubic splines have essentially the minimum regularity for handling certain edge cases; see Appendix A for the technical details.

Let $f_\theta \colon \mathbb{R}^w \to \mathbb{R}^{w \times (v+1)}$ be any neural network model depending on parameters $\theta$. The value $w$ is a hyperparameter describing the size of the hidden state. Let $\zeta_\theta \colon \mathbb{R}^{v+1} \to \mathbb{R}^w$ be any neural network model depending on parameters $\theta$.

Then we define the *neural controlled differential equation* model as the solution of the CDE

$$z_t = z_{t_0} + \int_{t_0}^{t} f_\theta(z_s) \mathrm{d}X_s \quad \text{for } t \in (t_0, t_n], \tag{3}$$

where $z_{t_0} = \zeta_\theta(x_0, t_0)$. This initial condition is used to avoid translational invariance. Analogous to RNNs, the output of the model may either be taken to be the evolving process $z$, or the terminal value $z_{t_n}$, and the final prediction should typically be given by a linear map applied to this output.

The resemblance between equations (1) and (3) is clear. The essential difference is that equation (3) is driven by the data process $X$, whilst equation (1) is driven only by the identity function $\iota \colon \mathbb{R} \to \mathbb{R}$. In this way, the Neural CDE is naturally adapting to incoming data, as changes in $X$ change the local dynamics of the system. See Figure 1.

### 3.1 Universal Approximation

It is a famous theorem in CDEs that in some sense they represent general functions on streams [22, Theorem 4.2], [23, Proposition A.6]. This may be applied to show that Neural CDEs are universal approximators, which we summarise in the following informal statement.

**Theorem (Informal).** *The action of a linear map on the terminal value of a Neural CDE is a universal approximator from* {sequences in $\mathbb{R}^v$} *to* $\mathbb{R}$.

Theorem B.14 in Appendix B gives a formal statement and a proof, which is somewhat technical. The essential idea is that CDEs may be used to approximate bases of functions on path space.

### 3.2 Evaluating the Neural CDE model

Evaluating the Neural CDE model is straightforward. In our formulation above, $X$ is in fact not just of bounded variation but is differentiable. In this case, we may define

$$g_{\theta, X}(z, s) = f_\theta(z) \frac{\mathrm{d}X}{\mathrm{d}s}(s), \tag{4}$$

so that for $t \in (t_0, t_n]$,

$$z_t = z_{t_0} + \int_{t_0}^t f_\theta(z_s)\mathrm{d}X_s = z_{t_0} + \int_{t_0}^t f_\theta(z_s)\frac{\mathrm{d}X}{\mathrm{d}s}(s)\mathrm{d}s = z_{t_0} + \int_{t_0}^t g_{\theta,X}(z_s, s)\mathrm{d}s. \qquad (5)$$

Thus it is possible to solve the Neural CDE using the same techniques as for Neural ODEs. In our experiments, we were able to straightforwardly use the already-existing `torchdiffeq` package [24] without modification.

### 3.3 Comparison to alternative ODE models

For the reader not familiar with CDEs, it might instead seem more natural to replace $g_{\theta,X}$ with some $h_\theta(z, X_s)$ that is directly applied to and potentially nonlinear in $X_s$. Indeed, such approaches have been suggested before, in particular to derive a "GRU-ODE" analogous to a GRU [14, 25].

However, it turns out that something is lost by doing so, which we summarise in the following statement.

**Theorem (Informal).** *Any equation of the form $z_t = z_0 + \int_{t_0}^t h_\theta(z_s, X_s)\mathrm{d}s$ may be represented exactly by a Neural CDE of the form $z_t = z_0 + \int_{t_0}^t f_\theta(z_s)\mathrm{d}X_s$. However the converse statement is not true.*

Theorem C.1 in Appendix C provides the formal statement and proof. The essential idea is that a Neural CDE can easily represent the identity function between paths, whilst the alternative cannot.

In our experiments, we find that the Neural CDE substantially outperforms the GRU-ODE, which we speculate is a consequence of this result.

### 3.4 Training via the adjoint method

An attractive part of Neural ODEs is the ability to train via adjoint backpropagation, see [15, 26, 27, 28], which uses only $\mathcal{O}(H)$ memory in the time horizon $L = t_n - t_0$ and the memory footprint $H$ of the vector field. This is contrast to directly backpropagating through the operations of an ODE solver, which requires $\mathcal{O}(LH)$ memory.

Previous work on Neural ODEs for time series, for example [13], has interrupted the ODE to make updates at each observation. Adjoint-based backpropagation cannot be performed across the jump, so this once again requires $\mathcal{O}(LH)$ memory.

In contrast, the $g_{\theta,X}$ defined by equation (4) continuously incorporates incoming data, without interrupting the differential equation, and so adjoint backpropagation may be performed. This requires only $\mathcal{O}(H)$ memory. The underlying data unavoidably uses an additional $\mathcal{O}(L)$ memory. Thus training the Neural CDE has an overall memory footprint of just $\mathcal{O}(L + H)$.

We do remark that the adjoint method should be used with care, as some systems are not stable to evaluate in both the forward and backward directions [29, 30]. The problem of finite-time blow-up is at least not a concern, given global Lipschitz conditions on the vector field [20, Theorem 1.3]. Such a condition will be satisfied if $f_\theta$ uses ReLU or tanh nonlinearities, for example.

### 3.5 Intensity as a channel

It has been observed that the frequency of observations may carry information [6]. For example, doctors may take more frequent measurements of patients they believe to be at greater risk. Some previous work has for example incorporated this information by learning an intensity function [12, 13, 15].

We instead present a simple *non-learnt* procedure, that is compatible with Neural CDEs. Simply concatenate the index $i$ of $x_i$ together with $x_i$, and then construct a path $X$ from the pair $(i, x_i)$, as before. The channel of $X$ corresponding to these indices then corresponds to the *cumulative* intensity of observations.

As the derivative of $X$ is what is then used when evaluating the Neural CDE model, as in equation (5), then it is the intensity itself that then determines the vector field.

### 3.6 Partially observed data

One advantage of our formulation is that it naturally adapts to the case of partially observed data. Each channel may independently be interpolated between observations to define $X$ in exactly the same manner as before.

In this case, the procedure for measuring observational intensity in Section 3.5 may be adjusted by instead having a separate observational intensity channel $c_i$ for each original channel $o_i$, such that $c_i$ increments every time an observation is made in $o_i$.

### 3.7 Batching

Given a batch of training samples with observation times drawn from the same interval $[t_0, t_n]$, we may interpolate each $\mathbf{x}$ to produce a continuous $X$, as already described. Each path $X$ is what may then be batched together, regardless of whether the underlying data is irregularly sampled or partially observed. Batching is thus efficient for the Neural CDE model.

## 4 Experiments

We benchmark the Neural CDE against a variety of existing models.

These are: GRU-$\Delta$t, which is a GRU with the time difference between observations additionally used as an input; GRU-D [6], which modifies the GRU-$\Delta$t with learnt exponential decays between observations; GRU-ODE [14, 25], which is an ODE analogous to the operation of a GRU and uses $X$ as its input; ODE-RNN [13], which is a GRU-$\Delta$t model which additionally applies a learnt Neural ODE to the hidden state between observations. Every model then used a learnt linear map from the final hidden state to the output, and was trained with cross entropy or binary cross entropy loss.

The GRU-$\Delta$t represents a straightforward baseline, the GRU-ODE is an alternative ODE model that is thematically similar to a Neural CDE, and the GRU-D and ODE-RNNs are state-of-the-art models for these types of problems. To avoid unreasonably extensive comparisons we have chosen to focus on demonstrating superiority within the class of ODE and RNN based models to which the Neural CDE belongs. These models were selected to collectively be representative of this class.

Each model is run five times, and we report the mean and standard deviation of the test metrics.

For every problem, the hyperparameters were chosen by performing a grid search to optimise the performance of the baseline ODE-RNN model. Equivalent hyperparameters were then used for every other model, adjusted slightly so that every model has a comparable number of parameters.

Precise experimental details may be found in Appendix D, regarding normalisation, architectures, activation functions, optimisation, hyperparameters, regularisation, and so on.

### 4.1 Varying amounts of missing data on CharacterTrajectories

We begin by demonstrating the efficacy of Neural CDEs on irregularly sampled time series.

To do this, we consider the CharacterTrajectories dataset from the UEA time series classification archive [31]. This is a dataset of 2858 time series, each of length 182, consisting of the $x, y$ position and pen tip force whilst writing a Latin alphabet character in a single stroke. The goal is to classify which of 20 different characters are written.

We run three experiments, in which we drop either 30%, 50% or 70% of the data. The observations to drop are selected uniformly at random and independently for each time series. Observations are removed across channels, so that the resulting dataset is irregularly sampled but completely observed. The randomly removed data is the same for every model and every repeat.

The results are shown in Table 1. The Neural CDE outperforms every other model considered, and furthermore it does so whilst using an order of magnitude less memory. The GRU-ODE does consistently poorly despite being the most theoretically similar model to a Neural CDE. Furthermore we see that even as the fraction of dropped data increases, the performance of the Neural CDE remains roughly constant, whilst the other models all start to decrease.

Further experimental details may be found in Appendix D.2.

Table 1: Test accuracy (mean ± std, computed across five runs) and memory usage on CharacterTrajectories. Memory usage is independent of repeats and of amount of data dropped.

| Model | Test Accuracy | | | Memory usage (MB) |
|---|---|---|---|---|
| | 30% dropped | 50% dropped | 70% dropped | |
| GRU-ODE | 92.6% ± 1.6% | 86.7% ± 3.9% | 89.9% ± 3.7% | 1.5 |
| GRU-$\Delta$t | 93.6% ± 2.0% | 91.3% ± 2.1% | 90.4% ± 0.8% | 15.8 |
| GRU-D | 94.2% ± 2.1% | 90.2% ± 4.8% | 91.9% ± 1.7% | 17.0 |
| ODE-RNN | 95.4% ± 0.6% | 96.0% ± 0.3% | 95.3% ± 0.6% | 14.8 |
| Neural CDE (ours) | **98.7% ± 0.8%** | **98.8% ± 0.2%** | **98.6% ± 0.4%** | **1.3** |

Table 2: Test AUC (mean ± std, computed across five runs) and memory usage on PhysioNet sepsis prediction. 'OI' refers to the inclusion of observational intensity, 'No OI' means without it. Memory usage is independent of repeats.

| Model | Test AUC | | Memory usage (MB) | |
|---|---|---|---|---|
| | OI | No OI | OI | No OI |
| GRU-ODE | 0.852 ± 0.010 | 0.771 ± 0.024 | 454 | 273 |
| GRU-$\Delta$t | 0.878 ± 0.006 | 0.840 ± 0.007 | 837 | 826 |
| GRU-D | 0.871 ± 0.022 | **0.850 ± 0.013** | 889 | 878 |
| ODE-RNN | 0.874 ± 0.016 | 0.833 ± 0.020 | 696 | 686 |
| Neural CDE (ours) | **0.880 ± 0.006** | 0.776 ± 0.009 | **244** | **122** |

## 4.2 Observational intensity with PhysioNet sepsis prediction

Next we consider a dataset that is both irregularly sampled and partially observed, and investigate the benefits of observational intensity as discussed in Sections 3.5 and 3.6.

We use data from the PhysioNet 2019 challenge on sepsis prediction [32, 33]. This is a dataset of 40335 time series of variable length, describing the stay of patients within an ICU. Measurements are made of 5 static features such as age, and 34 time-dependent features such as respiration rate or creatinine concentration in the blood, down to an hourly resolution. Most values are missing; only 10.3% of values are observed. We consider the first 72 hours of a patient's stay, and consider the binary classification problem of predicting whether they develop sepsis over the course of their entire stay (which is as long as a month for some patients).

We run two experiments, one with observational intensity, and one without. For the Neural CDE and GRU-ODE models, observational intensity is continuous and on a per-channel basis as described in Section 3.6. For the ODE-RNN, GRU-D, and GRU-$\Delta$t models, observational intensity is given by appending an observed/not-observed mask to the input at each observation.[2][3] The initial hidden state of every model is taken to be a function (a small single hidden layer neural network) of the static features.

The results are shown in Table 2. As the dataset is highly imbalanced (5% positive rate), we report AUC rather than accuracy. When observational intensity is used, then the Neural CDE produces the best AUC overall, although the ODE-RNN and GRU-$\Delta$t models both perform well. The GRU-ODE continues to perform poorly.

Without observational intensity then every model performs substantially worse, and in particular we see that the benefit of including observational intensity is particularly dramatic for the Neural CDE.

As before, the Neural CDE remains the most memory-efficient model considered.

Further experimental details can be found in Appendix D.3.

## 4.3 Regular time series with Speech Commands

Finally we demonstrate the efficacy of Neural CDE models on regularly spaced, fully observed time series, where we might hypothesise that the baseline models will do better.

We used the Speech Commands dataset [34]. This consists of one-second audio recordings of both background noise and spoken words such as 'left', 'right', and so on. We used 34975 time series corresponding to ten spoken words so as to produce a balanced classification problem. We preprocess the dataset by computing mel-frequency cepstrum coefficients so that each time series is then regularly spaced with length 161 and 20 channels.

Table 3: Test Accuracy (mean $\pm$ std, computed across five runs) and memory usage on Speech Commands. Memory usage is independent of repeats.

| Model | Test Accuracy | Memory usage (GB) |
|---|---|---|
| GRU-ODE | $47.9\% \pm 2.9\%$ | **0.164** |
| GRU-$\Delta$t | $43.3\% \pm 33.9\%$ | 1.54 |
| GRU-D | $32.4\% \pm 34.8\%$ | 1.64 |
| ODE-RNN | $65.9\% \pm 35.6\%$ | 1.40 |
| Neural CDE (ours) | **89.8%** $\pm$ **2.5%** | 0.167 |

The results are shown in Table 3. We observed that the Neural CDE had the highest performance, whilst using very little memory. The GRU-ODE consistently failed to perform. The other benchmark models surprised us by exhibiting a large variance on this problem, due to sometimes failing to train, and we were unable to resolve this by tweaking the optimiser. The best GRU-$\Delta$t, GRU-D and ODE-RNN models did match the performance of the Neural CDE, suggesting that on a regularly spaced problem all approaches can be made to work equally well.

In contrast, the Neural CDE model produced consistently good results every time. Anecdotally this aligns with what we observed over the course of all of our experiments, which is that the Neural CDE model usually trained quickly, and was robust to choice of optimisation hyperparameters. We stress that we did not perform a formal investigation of this phenomenen.

Further experimental details can be found in Appendix D.4.

## 5 Related work

In [13, 14] the authors consider interrupting a Neural ODE with updates from a recurrent cell at each observation, and were in fact the inspiration for this paper. Earlier work [6, 7, 8, 9] use intra-observation exponential decays, which are a special case. [35] consider something similar by interrupting a Neural ODE with stochastic events.

SDEs and CDEs are closely related, and several authors have introduced Neural SDEs. [36, 37, 38] treat them as generative models for time series and seek to model the data distribution. [39, 40] investigate using stochasticity as a regularizer, and demonstrate better performance by doing so. [41] use random vector fields so as to promote simpler trajectories, but do not use the 'SDE' terminology.

Adjoint backpropagation needs some work to apply to SDEs, and so [42, 43, 44] all propose methods for training Neural SDEs. We would particularly like to highlight the elegant approach of [44], who use the pathwise treatment given by rough analysis to approximate Brownian noise, and thus produce a random Neural ODE which may be trained in the usual way; such approaches may also avoid the poor convergence rates of SDE solvers as compared to ODE solvers.

Other elements of the theory of rough analysis and CDEs have also found machine learning applications. Amongst others, [23, 45, 46, 47, 48, 49, 50, 51] discuss applications of the signature transform to time series problems, and [52] investigate the related logsignature transform. [53] develop a kernel for time series using this methodology, and [54] apply this kernel to Gaussian processes. [55] develop software for these approaches tailored for machine learning.

There has been a range of work seeking to improve Neural ODEs. [56, 57] investigate speed-ups to the training proecedure, [58] develop an energy-based Neural ODE framework, and [29] demonstrate potential pitfalls with adjoint backpropagation. [30, 59] consider ways to vary the network parameters over time. [57, 60] consider how a Neural ODE model may be regularised (see also the stochastic regularisation discussed above). This provides a wide variety of techniques, and we are hopeful that some of them may additionally carry over to the Neural CDE case.

# 6 Discussion

## 6.1 Considerations

There are two key elements of the Neural CDE construction which are subtle, but important.

**Time as a channel**     CDEs exhibit a *tree-like invariance* property [18]. What this means, roughly, is that a CDE is blind to speed at which $X$ is traversed. Thus merely setting $X_{t_i} = x_i$ would not be enough, as time information is only incorporated via the parameterisation. This is why time is explicitly included as a channel via $X_{t_i} = (x_i, t_i)$.

**Initial value networks**     The initial hidden state $z_{t_0}$ should depend on $X_{t_0} = (x_0, t_0)$. Otherwise, the Neural CDE will depend upon $X$ only through its derivative $\mathrm{d}X/\mathrm{d}t$, and so will be translationally invariant. An alternative would be to append another channel whose first derivative includes translation-sensitive information, for example by setting $X_{t_i} = (x_i, t_i, t_i x_0)$.

## 6.2 Performance tricks

We make certain (somewhat anecdotal) observations of tricks that seemed to help performance.

**Final tanh nonlinearity**     We found it beneficial to use a tanh as a final nonlinearity for the vector field $f_\theta$ of a Neural CDE model. Doing so helps prevent extremely large initial losses, as the tanh constrains the rate of change of the hidden state. This is analogous to RNNs, where the key feature of GRUs and LSTMs are procedures to constrain the rate of change of the hidden state.

**Layer-wise learning rates**     We found it beneficial to use a larger ($\times 10$–100) learning rate for the linear layer on top of the output of the Neural CDE, than for the vector field $f_\theta$ of the Neural CDE itself. This was inspired by the observation that the final linear layer has (in isolation) only a convex optimisation problem to solve.[4]

## 6.3 Limitations

**Speed of computation**     We found that Neural CDEs were typically slightly faster to compute than the ODE-RNN model of [13]. (This is likely to be because in an Neural CDE, steps of the numerical solver can be made across observations, whilst the ODE-RNN must interrupt its solve at each observation.)

However, Neural CDEs were still roughly fives times slower than RNN models. We believe this is largely an implementation issue, as the implementation via `torchdiffeq` is in Python, and by default uses double-precision arithmetic with variable step size solvers, which we suspect is unnecessary for most practical tasks.

**Number of parameters**     If the vector field $f_\theta \colon \mathbb{R}^w \to \mathbb{R}^{w \times (v+1)}$ is a feedforward neural network, with final hidden layer of size $\omega$, then the number of scalars for the final affine transformation is of size $\mathcal{O}(\omega v w)$, which can easily be very large. In our experiments we have to choose small values of $w$ and $\omega$ for the Neural CDE to ensure that the number of parameters is the same across models.

We did experiment with representing the final linear layer as an outer product of transformations $\mathbb{R}^w \to \mathbb{R}^w$ and $\mathbb{R}^w \to \mathbb{R}^{v+1}$. This implies that the resulting matrix is rank-one, and reduces the number of parameters to just $\mathcal{O}(\omega(v+w))$, but unfortunately we found that this hindered the classification performance of the model.

## 6.4 Future work

**Vector field design**     The vector fields $f_\theta$ that we consider are feedforward networks. More sophisticated choices may allow for improved performance, in particular to overcome the trilinearity issue just discussed.

**Modelling uncertainty**    As presented here, Neural CDEs do not give any measure of uncertainty about their predictions. Such extensions are likely to be possible, given the close links between CDEs and SDEs, and existing work on Neural SDEs [36, 37, 38, 39, 40, 42, 43, 44, 61].

**Numerical schemes**    In this paper we integrated the Neural CDE by reducing it to an ODE. The field of numerical CDEs is relatively small – to the best of our knowledge [17, 62, 63, 64, 65, 66, 67, 68, 69] constitute essentially the entire field, and are largely restricted to rough controls. Other numerical methods may be able to exploit the CDE structure to improve performance.

**Choice of $X$**    Natural cubic splines were used to construct the path $X$ from the time series **x**. However, these are not causal. That is, $X_t$ depends upon the value of $x_i$ for $t < t_i$. This makes it infeasible to apply Neural CDEs in real-time settings, for which $X_t$ is needed before $x_i$ has been observed. Resolving this particular issue is a topic on which we have follow-up work planned.

**Other problem types**    Our experiments here involved only classification problems. There was no real reason for this choice, and we expect Neural CDEs to be applicable more broadly.

## 6.5   Related theories

**Rough path theory**    The field of rough path theory, which deals with the study of CDEs, is much larger than the small slice that we have used here. It is likely that further applications may serve to improve Neural CDEs. A particular focus of rough path theory is how to treat functions that must be sensitive to the order of events in a particular (continuous) way.

**Control theory**    Despite their similar names, and consideration of similar-looking problems, control theory and controlled differential equations are essentially separate fields. Control theory has clear links and applications that may prove beneficial to models of this type.

**RNN theory**    Neural CDEs may be interpreted as continuous-time versions of RNNs. CDEs thus offer a theoretical construction through which RNNs may perhaps be better understood. Conversely, what is known about RNNs may have applications to improve Neural CDEs.

## 7   Conclusion

We have introduced a new class of continuous-time time series models, Neural CDEs. Just as Neural ODEs are the continuous analogue of ResNets, the Neural CDE is the continuous time analogue of an RNN. The model has three key advantages: it operates directly on irregularly sampled and partially observed multivariate time series, it demonstrates state-of-the-art performance, and it benefits from memory-efficient adjoint-based backpropagation even across observations. To the best of our knowledge, no other model combines these three features together. We also provide additional theoretical results demonstrating universal approximation, and that Neural CDEs subsume alternative ODE models.

## Broader Impact

We have introduced a new tool for studying irregular time series. As with any tool, it may be used in both positive and negative ways. The authors have a particular interest in electronic health records (an important example of irregularly sampled time-stamped data) and so here at least we hope and expect to see a positive impact from this work. We do not expect any specific negative impacts from this work.

## Acknowledgments and Disclosure of Funding

Thanks to Cristopher Salvi for many vigorous discussions on this topic. PK was supported by the EPSRC grant EP/L015811/1. JM was supported by the EPSRC grant EP/L015803/1 in collaboration with Iterex Therapuetics. JF was supported by the EPSRC grant EP/N509711/1. PK, JM, JF, TL were supported by the Alan Turing Institute under the EPSRC grant EP/N510129/1.

## Footnotes

[1]Not to be confused with the similarly-named but separate field of control theory.

[2]As our proposed observational intensity goes via a derivative, these each contain the same information.

[3]Note that the ODE-RNN, GRU-D and GRU-$\Delta$t models always receive the time difference between observations, $\Delta$t, as an input. Thus even in the no observational intensity case, they remain aware of the irregular sampling of the data, and so this case not completely fair to the Neural CDE and GRU-ODE models.

[4]In our experiments we applied this learning rate to the linear layer on top of every model, not the just the Neural CDE, to ensure a fair comparison.

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
