[Supplementary Material]

# Supplementary material

Appendix A discusses the technical considerations of schemes for constructing a path $X$ from data. Appendix B proves universal approximation of the Neural CDE model, and is substantially more technical than the rest of this paper. Appendix C proves that the Neural CDE model subsumes alternative ODE models which depend directly and nonlinearly on the data. Appendix D gives the full details of every experiment, such as choice of optimiser, hyperparameter searches, and so on.

## A   Other schemes for constructing the path $X$

To evaluate the model as discussed in Section 3.2, $X$ must be at least continuous and piecewise differentiable.

### A.1   Differentiating with respect to the time points

However, there is a technical caveat in the specific case that derivatives with respect to the initial time $t_0$ are required, and that training is done with the adjoint method. In this case the derivative with respect to $t_0$ is computed using, and thus requires, derivatives of the vector field with respect to $t$.

To be precise, suppose we have a Neural CDE as before:

$$z_t = z_{t_0} + \int_{t_0}^t f_\theta(z_s)\mathrm{d}X_s \quad \text{for } t \in (t_0, t_n].$$

Let $L$ be some (for simplicity scalar-valued) function of $z_{t_n}$, for example a loss. Consider

$$g_{\theta,X}(z,s) = f_\theta(z)\frac{\mathrm{d}X}{\mathrm{d}s}(s)$$

as before, and let

$$a_s = \frac{\mathrm{d}L}{\mathrm{d}z_s},$$

which is vector-valued, with size equal to the size of $z_s$, the number of hidden channels.

Then, applying [15, Equation 52] to our case:

$$\begin{aligned}
\frac{\mathrm{d}L}{\mathrm{d}t_0} &= \frac{\mathrm{d}L}{\mathrm{d}t_n} - \int_{t_n}^{t_0} a_s \cdot \frac{\partial g_{\theta,X}}{\partial s}(z_s, s)\mathrm{d}s \\
&= \frac{\mathrm{d}L}{\mathrm{d}t_n} - \int_{t_n}^{t_0} a_s \cdot f_\theta(z_s)\frac{\mathrm{d}^2 X}{\mathrm{d}s^2}(s)\mathrm{d}s,
\end{aligned} \tag{6}$$

where $\cdot$ represents the dot product.

In principle we may make sense of equation (6) when $\mathrm{d}^2X/\mathrm{d}s^2$ is merely measure valued, but in practice most code is only set up to handle classical derivatives. If derivatives with respect to $t_0$ are desired, then practically speaking $X$ must be at least twice differentiable.

### A.2   Adaptive step size solvers

There is one further caveat that must be considered. Suppose $X$ is twice differentiable, but that the second derivative is discontinuous. For example this would be accomplished by taking $X$ to be a quadratic spline interpolation.

If seeking to solve equation (6) with an adaptive step-size solver, we found that the solver would take a long time to compute the backward pass, as it would have to slow down to resolve each jump in $\mathrm{d}^2X/\mathrm{d}s^2$, and then speed back up in the intervals in-between.

### A.3 Natural cubic splines

This is then the reason for our selection of natural cubic splines: by ensuring that $X$ is twice continuously differentiable, the above issue is ameliorated, and adaptive step size solvers operate acceptably. Cubic splines give essentially the minimum regularity for the techniques discussed in this paper to work 'out of the box' in all cases.

Other than this smoothness, however, there is little that is special about natural cubic splines. Other possible options are for example Gaussian processes [10, 11] or kernel methods [12]. Furthermore, especially in the case of noisy data it need not be an interpolation scheme – approximation and curve-fitting schemes are valid too.

## B  Universal Approximation

The behaviour of controlled differential equations are typically described through the *signature transform* (also known as *path signature* or simply *signature*) [20] of the driving process $X$. We demonstrate here how a (Neural) CDE may be reduced to consideration of just the signature transform, in order to prove universal approximation.

The proof is essentially split into two parts. The first part is to prove universal approximation with respect to continuous paths, as is typically done for CDEs. The second (lengthier) part is to interpret what this means for the natural cubic splines that we use in this paper, so that we can get universal approximation with respect to the original data as well.

**Definition B.1.** Let $\tau, T \in \mathbb{R}$ with $\tau < T$ and let $v \in \mathbb{N}$. Let $\mathcal{V}^1([\tau, T]; \mathbb{R}^v)$ represent the space of continuous functions of bounded variation. Equip this space with the norm

$$\widehat{X} \mapsto \left\| \widehat{X} \right\|_{\mathcal{V}} = \left\| \widehat{X} \right\|_\infty + \left| \widehat{X} \right|_{BV}.$$

This is a somewhat unusual norm to use, as bounded variation seminorms are more closely aligned with $L^1$ norms than $L^\infty$ norms.

**Definition B.2.** For any $\widehat{X} \in \mathcal{V}^1([\tau, T]; \mathbb{R}^v)$ let $X_t = (\widehat{X}_t, t) \in \mathcal{V}^1([\tau, T]; \mathbb{R}^{v+1})$. We choose to use the notation of 'removing the hat' to denote this time augmentation, for consistency with the main text which uses $X$ for the time-augmented path.

**Definition B.3.** For any $N, v \in \mathbb{N}$, let $\kappa(N, v) = \sum_{i=0}^{N}(v+1)^i$.

**Definition B.4** (Signature transform). For any $k \in \mathbb{N}$ and any $y \in \mathbb{R}^k$, let $M(y) \in \mathbb{R}^{k(v+1) \times (v+1)}$ be the matrix

$$M(y) = \begin{bmatrix} y^1 & 0 & 0 & \cdots & 0 \\ y^2 & 0 & 0 & \cdots & 0 \\ \vdots & \vdots & \vdots & & \vdots \\ y^k & 0 & 0 & \cdots & 0 \\ 0 & y^1 & 0 & \cdots & 0 \\ 0 & \vdots & 0 & & \vdots \\ 0 & y^k & 0 & \cdots & 0 \\ & & & \ddots & \\ 0 & 0 & 0 & \cdots & y^1 \\ \vdots & \vdots & \vdots & \cdots & \vdots \\ 0 & 0 & 0 & \cdots & y^k \end{bmatrix}$$

Fix $N \in \mathbb{N}$ and $X \in \mathcal{V}^1([\tau, T]; \mathbb{R}^{v+1})$. Let $y^{0,X,N} : [\tau, T] \to \mathbb{R}$ be constant with $y_t^{0,X,N} = 1$.

For all $i \in \{1, \ldots, N\}$, iteratively let $y^{i,X,N} : [\tau, T] \to \mathbb{R}^{(v+1)^i}$ be the value of the integral

$$y_t^{i,X,N} = y_\tau^{i,X,N} + \int_\tau^t M(y_s^{i-1,X,N}) \mathrm{d}X_s \quad \text{for } t \in (\tau, T],$$

with $y_\tau^{i,X,N} = 0 \in \mathbb{R}^{(v+1)^i}$.

Then we may stack these together:

$$y^{X,N} = (y^{0,X,N}, \ldots, y^{N,X,N}) \colon [\tau, T] \to \mathbb{R}^{\kappa(N,v)},$$
$$\widetilde{M}(y^{X,N}) = (0, M \circ y^{0,X,N}, \ldots, M \circ y^{N-1,X,N}) \colon [\tau, T] \to \mathbb{R}^{\kappa(N,v) \times v}.$$

Then $y^{X,N}$ is the unique solution to the CDE

$$y_t^{X,N} = y_\tau^{X,N} + \int_\tau^t \widetilde{M}(y^{X,N})_s \mathrm{d}X_s \quad \text{for } t \in (\tau, T],$$

with $y_\tau^{X,N} = (1, 0, \ldots, 0)$.

Then the *signature transform truncated to depth $N$* is defined as the map

$$\mathrm{Sig}^N \colon \mathcal{V}^1([\tau, T]; \mathbb{R}^{v+1}) \to \mathbb{R}^{\kappa(N,v)},$$
$$\mathrm{Sig}^N \colon X \mapsto y_T^{X,N}.$$

If this seems like a strange definition, then note that for any $a \in \mathbb{R}^k$ and any $b \in \mathbb{R}^v$ that $M(a)b$ is equal to a flattened vector corresponding to the outer product $a \otimes b$. As such, the signature CDE is instead typically written much more concisely as the exponential differential equation

$$y_t^{X,N} = y_\tau^{X,N} + \int_\tau^t y_s^{X,N} \otimes \mathrm{d}X_s \quad \text{for } t \in (\tau, T],$$

however we provide the above presentation for consistency with the rest of the text, which does not introduce $\otimes$.

**Definition B.5.** Let $\mathcal{V}_0^1([\tau, T]; \mathbb{R}^v) = \left\{ X \in \mathcal{V}^1([\tau, T]; \mathbb{R}^v) \,\middle|\, X_0 = 0 \right\}.$

With these definitions out of the way, we are ready to state the famous universal nonlinearity property of the signature transform. We think [22, Theorem 4.2] gives the most straightforward proof of this result. This essentially states that the signature gives a basis for the space of functions on compact path space.

**Theorem B.6** (Universal nonlinearity). *Let $\tau, T \in \mathbb{R}$ with $\tau < T$ and let $v, u \in \mathbb{N}$.*

*Let $K \subseteq \mathcal{V}_0^1([\tau, T]; \mathbb{R}^v)$ be compact. (Note the subscript zero.)*

*Let $\mathrm{Sig}^N \colon \mathcal{V}^1([\tau, T]; \mathbb{R}^{v+1}) \to \mathbb{R}^{\kappa(N,v)}$ denote the signature transform truncated to depth $N$.*

*Let*

$$J^{N,u} = \left\{ \ell \colon \mathbb{R}^{\kappa(N,v)} \to \mathbb{R}^u \,\middle|\, \ell \text{ is linear} \right\}.$$

*Then*

$$\bigcup_{N \in \mathbb{N}} \left\{ \widehat{X} \mapsto \ell(\mathrm{Sig}^N(X)) \,\middle|\, \ell \in J^{N,u} \right\}$$

*is dense in $C(K; \mathbb{R}^u)$.*

With the universal nonlinearity property, we can now prove universal approximation of CDEs with respect to controlling paths $X$.

**Theorem B.7** (Universal approximation with CDEs). *Let $\tau, T \in \mathbb{R}$ with $\tau < T$ and let $v, u \in \mathbb{N}$. For any $w \in \mathbb{N}$ let*

$$F^w = \left\{ f \colon \mathbb{R}^w \to \mathbb{R}^{w \times (v+1)} \,\middle|\, f \text{ is continuous} \right\},$$
$$L^{w,u} = \left\{ \ell \colon \mathbb{R}^w \to \mathbb{R}^u \,\middle|\, \ell \text{ is linear} \right\},$$
$$\xi^w = \left\{ \zeta \colon \mathbb{R}^{v+1} \to \mathbb{R}^w \,\middle|\, \zeta \text{ is continuous} \right\}.$$

*For any $w \in \mathbb{N}$, any $f \in F^w$ and any $\zeta \in \xi^w$ and any $\widehat{X} \in \mathcal{V}^1([\tau, T]; \mathbb{R}^v)$, let $z^{f,\zeta,X} \colon [\tau, T] \to \mathbb{R}^w$ be the unique solution to the CDE*

$$z_t^{f,\zeta,X} = z_\tau^{f,\zeta,X} + \int_\tau^t f(z_s^{f,\zeta,X}) \mathrm{d}X_s \quad \text{for } t \in (\tau, T],$$

*with* $z_\tau^{f,\varsigma,X} = \varsigma(X_\tau)$.

*Let $K \subseteq \mathcal{V}^1([\tau, T]; \mathbb{R}^v)$ be compact.*

*Then*

$$\bigcup_{w \in \mathbb{N}} \left\{ \widehat{X} \mapsto \ell(z_T^{f,\varsigma,X}) \;\middle|\; f \in F^w, \ell \in L^{w,u}, \varsigma \in \xi^w \right\}$$

*is dense in $C(K; \mathbb{R}^u)$.*

*Proof.* We begin by prepending a straight line segment to every element of $K$. For every $\widehat{X} \in K$, define $\widehat{X}^* \colon [\tau - 1, T] \to \mathbb{R}^v$ by

$$\widehat{X}_t^* = \begin{cases} (t - \tau + 1)\widehat{X}_\tau & t \in [\tau - 1, \tau), \\ \widehat{X}_t & t \in [\tau, T]. \end{cases}$$

Similarly define $X^*$, so that a hat means that time is *not* a channel, whilst a star means that an extra straight-line segment has been prepended to the path.

Now let $K^* = \left\{ \widehat{X}^* \;\middle|\; \widehat{X} \in K \right\}$. Then $K^* \subseteq \mathcal{V}_0^1([\tau - 1, T]; \mathbb{R}^v)$ and is also compact. Therefore by Theorem B.6,

$$\bigcup_{N \in \mathbb{N}} \left\{ \widehat{X}^* \mapsto \ell(\mathrm{Sig}^N(X^*)) \;\middle|\; \ell \in J^{N,u} \right\}$$

is dense in $C(K^*; \mathbb{R}^u)$.

So let $\alpha \in C(K; \mathbb{R}^u)$ and $\varepsilon > 0$. The map $\widehat{X} \mapsto \widehat{X}^*$ is a homeomorphism, so we may find $\beta \in C(K^*; \mathbb{R}^u)$ such that $\beta(\widehat{X}^*) = \alpha(\widehat{X})$ for all $\widehat{X} \in K$. Next, there exists some $N \in \mathbb{N}$ and $\ell \in J^{N,u}$ such that $\gamma$ defined by $\gamma \colon \widehat{X}^* \mapsto \ell(\mathrm{Sig}^N(X^*))$ is $\varepsilon$-close to $\beta$.

By Definition B.4 there exists $f \in F^{\kappa(N,v)}$ so that $\mathrm{Sig}^N(X^*) = y_T^{X^*}$ for all $X^* \in K^*$, where $y^{X^*}$ is the unique solution of the CDE

$$y_t^{X^*} = y_{\tau-1}^{X^*} + \int_{\tau-1}^t f(y_s^{X^*}) \mathrm{d}X_s^* \quad \text{for } t \in (\tau - 1, T],$$

with $y_{\tau-1}^{X^*} = (1, 0, \ldots, 0)$.

Now let $\varsigma \in \xi$ be defined by $\varsigma(X_\tau) = y_\tau^{X^*}$, which we note is well defined because the value of $y_t^{X^*}$ only depends on $X_\tau$ for $t \in [\tau - 1, \tau]$.

Now for any $X \in K$ let $z^X \colon [\tau, T] \to \mathbb{R}^w$ be the unique solution to the CDE

$$z_t^X = z_\tau^X + \int_\tau^t f(z_s^X) \mathrm{d}X_s \quad \text{for } t \in (\tau, T],$$

with $z_\tau^X = \varsigma(X_\tau)$.

Then by uniqueness of solution, $z_t^X = y_t^{X^*}$ for $t \in [\tau, T]$, and so in particular $\mathrm{Sig}^N(X^*) = y_T^{X^*} = z_T^X$.

Finally it remains to note that $\ell \in J^{N,u} = L^{\kappa(N,v),u}$.

So let $\delta$ be defined by $\delta \colon \widehat{X} \mapsto \ell(z_T^X)$. Then $\delta$ is in the set which we were aiming to show density of (with $w = \kappa(N, v)$, $f \in F^w$, $\ell \in L^{w,u}$ and $\varsigma \in \xi$ as chosen above), whilst for all $\widehat{X} \in K$,

$$\delta(\widehat{X}) = \ell(z_T^X) = \ell(\mathrm{Sig}^N(X^*)) = \gamma(\widehat{X}^*)$$

is $\varepsilon$-close to $\beta(\widehat{X}^*) = \alpha(\widehat{X})$. Thus density has been established. $\qquad\square$

**Lemma B.8.** *Let $K \subseteq C^2([\tau, T], \mathbb{R}^v)$ be uniformly bounded with uniformly bounded first and second derivatives. That is, there exists some $C > 0$ such that $\left\| \widehat{X} \right\|_\infty + \left\| \mathrm{d}\widehat{X}/\mathrm{d}t \right\|_\infty + \left\| \mathrm{d}^2\widehat{X}/\mathrm{d}t^2 \right\|_\infty < C$ for all $\widehat{X} \in K$. Then $K \subseteq \mathcal{V}^1([\tau, T]; \mathbb{R}^v)$ and is relatively compact (that is, its closure is compact) with respect to $\| \cdot \|_\mathcal{V}$.*

*Proof.* $K$ is bounded in $C^2([\tau, T], \mathbb{R}^v)$ so it is relatively compact in $C^1([\tau, T], \mathbb{R}^v)$.

Furthermore for any $\widehat{X} \in K$,

$$
\begin{aligned}
\left\|\widehat{X}\right\|_{\mathcal{V}} &= \left\|\widehat{X}\right\|_\infty + \left|\widehat{X}\right|_{BV} \\
&= \left\|\widehat{X}\right\|_\infty + \left\|\frac{\mathrm{d}\widehat{X}}{\mathrm{d}t}\right\|_1 \\
&\leq \left\|\widehat{X}\right\|_\infty + \left\|\frac{\mathrm{d}\widehat{X}}{\mathrm{d}t}\right\|_\infty,
\end{aligned}
$$

and so the embedding $C^1([\tau, T], \mathbb{R}^v) \to \mathcal{V}^1([\tau, T]; \mathbb{R}^v)$ is continuous. Therefore $K$ is also relatively compact in $\mathcal{V}^1([\tau, T]; \mathbb{R}^v)$. $\qquad\square$

Next we need to understand how a natural cubic spline is controlled by the size of its data. We establish the following crude bounds.

**Lemma B.9.** *Let $v \in \mathbb{N}$. Let $x_0, \ldots, x_n \in \mathbb{R}^v$. Let $t_0, \ldots, t_n \in \mathbb{R}$ be such that $t_0 < t_1 < \cdots < t_n$. Let $\widehat{X} \colon [t_0, t_n] \to \mathbb{R}^v$ be the natural cubic spline such that $\widehat{X}(t_i) = x_i$. Let $\tau_i = t_{i+1} - t_i$ for all $i$. Then there exists an absolute constant $C > 0$ such that*

$$
\left\|\widehat{X}\right\|_\infty + \left\|\mathrm{d}\widehat{X}/\mathrm{d}t\right\|_\infty + \left\|\mathrm{d}^2\widehat{X}/\mathrm{d}t^2\right\|_\infty < C \left\|\tau\right\|_\infty \left\|x\right\|_\infty (\min_i \tau_i)^{-2}(\left\|\tau\right\|_\infty + (\min_i \tau_i)^{-1}).
$$

*Proof.* Surprisingly, we could not find a reference for a fact of this type, but it follows essentially straightforwardly from the derivation of natural cubic splines.

Without loss of generality assume $v = 1$, as we are using the infinity norm over the dimensions $v$, and each cubic interpolation is performed separately for each dimension.

Let the $i$-th piece of $\widehat{X}$, which is a cubic on the interval $[t_i, t_{i+1}]$, be denoted $Y_i$. Without loss of generality, translate each $Y_i$ to the origin so as to simplify the algebra, so that $Y_i \colon [0, \tau_i] \to \mathbb{R}$. Let $Y_i(t) = a_i + b_i t + c_i t^2 + d_i t^3$ for some coefficients $a_i, b_i, c_i, d_i$ and $i \in \{0, \ldots, n-1\}$.

Letting $D_i = Y_i'(0)$ for $i \in \{0, \ldots, n-1\}$ and $D_n = Y_{n-1}'(\tau_{n-1})$, the displacement and derivative conditions imposed at each knot are $Y_i(0) = x_i$, $Y_i(\tau_i) = x_{i+1}$, $Y_i'(0) = D_i$ and $Y_i'(\tau_i) = D_{i+1}$. This then implies that $a_i = x_i$, $b_i = D_i$,

$$
c_i = 3\tau_i^{-2}(x_{i+1} - x_i) - \tau_i^{-1}(D_{i+1} + 2D_i), \tag{7}
$$

$$
d_i = 2\tau_i^{-3}(x_i - xi + 1) + \tau_i^{-2}(D_{i+1} + D_i). \tag{8}
$$

Letting $\lesssim$ denote 'less than or equal up to some absolute constant', then these equations imply that

$$
\left\|\widehat{X}\right\|_\infty = \max_i \|Y_i\|_\infty \lesssim \max_i(|x_i| + \tau_i |D_i|) \leq \|x\|_\infty + \|\tau\|_\infty \|D\|_\infty, \tag{9}
$$

$$
\left\|\frac{\mathrm{d}\widehat{X}}{\mathrm{d}t}\right\|_\infty = \max_i \|Y_i\|_\infty \lesssim \max_i(\tau_i^{-1}|x_i| + |D_i|) \leq \|x\|_\infty (\min_i \tau_i)^{-1} + \|D\|_\infty, \tag{10}
$$

$$
\left\|\frac{\mathrm{d}^2\widehat{X}}{\mathrm{d}t^2}\right\|_\infty = \max_i \|Y_i\|_\infty \lesssim \max_i(\tau_i^{-2}|x_i| + \tau_i^{-1}|D_i|) \leq \|x\|_\infty (\min_i \tau_i)^{-2} + \|D\|_\infty (\min_i \tau_i)^{-1}. \tag{11}
$$

Next, the second derivative condition at each knot is $Y_{i-1}''(\tau_{i-1}) = Y_i''(0)$ for $i \in \{1, \ldots, n-1\}$, and the natural condition is $Y_0''(0) = 0$ and $Y_{n-1}''(\tau_{n-1}) = 0$. With equations (7), (8) this gives

$$
\mathcal{T}D = k,
$$

where

$$\mathcal{T} = \begin{bmatrix} 2\tau_0^{-1} & \tau_0^{-1} \\ \tau_0^{-1} & 2(\tau_0^{-1}+\tau_1^{-1}) & \tau_1^{-1} \\ & \tau_1^{-1} & 2(\tau_1^{-1}+\tau_2^{-1}) & \tau_2^{-1} \\ & & \ddots & \ddots & \ddots \\ & & & \tau_{n-2}^{-1} & 2(\tau_{n-2}^{-1}+\tau_{n-1}^{-1}) & \tau_{n-1}^{-1} \\ & & & & \tau_{n-1}^{-1} & 2\tau_{n-1}^{-1} \end{bmatrix},$$

$$D = \begin{bmatrix} D_0 \\ \vdots \\ D_n \end{bmatrix},$$

$$k = \begin{bmatrix} 3\tau_0^{-2}(x_1-x_0) \\ 3\tau_1^{-2}(x_2-x_1) + 3\tau_0^{-2}(x_1-x_0) \\ \vdots \\ 3\tau_{n-1}^{-2}(x_n-x_{n-1}) + 3\tau_{n-2}^{-2}(x_{n-1}-x_{n-2}) \\ 3\tau_{n-1}^{-2}(x_n-x_{n-1}). \end{bmatrix}$$

Let $\left\|\mathcal{T}^{-1}\right\|_\infty$ denote the operator norm and $\|D\|_\infty$, $\|k\|_\infty$ denote the elementwise norm. Now $\mathcal{T}$ is diagonally dominant, so the Varah bound [70] and HM-AM inequality gives

$$\left\|\mathcal{T}^{-1}\right\|_\infty \le (\min_i(\tau_i^{-1}+\tau_{i+1}^{-1}))^{-1} \lesssim \|\tau\|_\infty.$$

Thus

$$\|D\|_\infty \lesssim \|\tau\|_\infty \|k\|_\infty \lesssim \|\tau\|_\infty \|x\|_\infty \, (\min_i \tau_i)^{-2}.$$

Together with equations (9)–(11) this gives the result. $\qquad\square$

**Definition B.10** (Space of time series)**.** Let $v \in \mathbb{N}$. and $\tau, T \in \mathbb{R}$ such that $\tau < T$. We define the space of time series in $[\tau, T]$ over $\mathbb{R}^v$ as

$$\mathcal{TS}_{[\tau,T]}(\mathbb{R}^v) = \{((t_0,x_0),\dots,(t_n,x_n)) \mid n \in \mathbb{N}, t_i \in [\tau,T], x_n \in \mathbb{R}^v, t_0 = \tau, t_n = T, n \ge 2\}.$$

To our knowledge, there is no standard topology on time series. One option is to treat them as sequences, however it is not clear how best to treat sequences of different lengths, or how to incorporate timestamp information. Given that a time series is typically some collection of observations from some underlying process, we believe the natural approach is to treat them as subspaces of functions.

**Definition B.11** (General topologies on time series)**.** Let $v \in \mathbb{N}$. and $\tau, T \in \mathbb{R}$ such that $\tau < T$. Let $F$ denote some topological space of functions. Let $\iota\colon \mathcal{TS}_{[\tau,T]}(\mathbb{R}^v) \to F$ be some map. Then we may define a topology on $\mathcal{TS}_{[\tau,T]}(\mathbb{R}^v)$ as the weakest topology with respect to which $\iota$ is continuous.

Recall that we use subscripts to denote function evaulation.

**Definition B.12** (Natural cubic spline topology)**.** Let $v \in \mathbb{N}$. and $\tau, T \in \mathbb{R}$ such that $\tau < T$. Let $F = C([\tau,T];\mathbb{R}^v)$ equipped with the uniform norm. For all $\mathbf{x} = ((t_0,x_0),\dots,(t_n,x_n)) \in \mathcal{TS}_{[\tau,T]}(\mathbb{R}^v)$, let $\widehat{\iota}\colon \mathcal{TS}_{[\tau,T]}(\mathbb{R}^v) \to F$ produce the natural cubic spline such that $\widehat{\iota}(\mathbf{x})_{t_i} = x_i$ with knots at $t_0,\dots,t_n$. Then this defines a topology on $\mathcal{TS}_{[\tau,T]}(\mathbb{R}^v)$ as in the previous definition.

**Remark B.13.** In fact this defines a seminorm on $\mathcal{TS}_{[\tau,T]}(\mathbb{R}^v)$, by $\|\mathbf{x}\| = \|\widehat{\iota}(\mathbf{x})\|_\infty$. This is only a seminorm as for example $((0,0),(2,2))$ and $((0,0),(1,1),(2,2))$ have the same natural cubic spline interpolation. This can be worked around so as to instead produce a full norm, but it is a deliberate choice not to: we would often prefer that these time series be thought of as equal. (And if it they are not equal, then first augmenting with observational intensity as in the main paper should distinguish them.)

**Theorem B.14** (Universal approximation with Neural CDEs via natural cubic splines). *Let* $\tau, T \in \mathbb{R}$ *with* $\tau < T$ *and let* $v, u \in \mathbb{N}$. *For any* $w \in \mathbb{N}$ *let*

$$F_{\mathcal{NN}}^w = \left\{ f \colon \mathbb{R}^w \to \mathbb{R}^{w \times (v+1)} \;\middle|\; f \text{ is a feedforward neural network} \right\},$$

$$L^{w,u} = \{ \ell \colon \mathbb{R}^w \to \mathbb{R}^u \mid \ell \text{ is linear} \},$$

$$\xi_{\mathcal{NN}}^w = \left\{ \zeta \colon \mathbb{R}^{v+1} \to \mathbb{R}^w \;\middle|\; \zeta \text{ is a feedforward neural network} \right\}.$$

*Let* $\widehat{\iota}$ *denote the natural cubic spline interpolation as in the previous definition, and recall that 'removing the hat' is our notation for augmenting with time. For any* $w \in \mathbb{N}$, *any* $f \in F^w$ *and any* $\zeta \in \xi_{\mathcal{NN}}^w$ *and any* $\mathbf{x} \in \mathcal{TS}_{[\tau,T]}(\mathbb{R}^v)$, *let* $z^{f,\zeta,\mathbf{x}} \colon [\tau, T] \to \mathbb{R}^w$ *be the unique solution to the CDE*

$$z_t^{f,\zeta,\mathbf{x}} = z_\tau^{f,\zeta,\mathbf{x}} + \int_\tau^t f(z_s^{f,\zeta,\mathbf{x}}) \mathrm{d}\iota(\mathbf{x})_s \quad \text{for } t \in (\tau, T],$$

*with* $z_\tau^{f,\zeta,\mathbf{x}} = \zeta(\iota(\mathbf{x})_\tau)$.

*Let* $K \subseteq \mathcal{TS}_{[\tau,T]}(\mathbb{R}^v)$ *be such that there exists* $C > 0$ *such that*

$$\|x\|_\infty \left( \min_i (t_{i+1} - t_i) \right)^{-3} < C \tag{12}$$

*for every* $\mathbf{x} = ((t_0, x_0), \ldots, (t_n, x_n)) \in K$. *(With* $C$ *independent of* $\mathbf{x}$.*)*

*Then*

$$\bigcup_{w \in \mathbb{N}} \left\{ \mathbf{x} \mapsto \ell(z_T^{f,\zeta,\mathbf{x}}) \;\middle|\; f \in F_{\mathcal{NN}}^w, \ell \in L^{w,u}, \zeta \in \xi_{\mathcal{NN}}^w \right\}$$

*is dense in* $C(K; \mathbb{R}^u)$ *with respect to the natural cubic spline topology on* $\mathcal{TS}_{[\tau,T]}(\mathbb{R}^v)$.

*Proof.* Fix $\mathbf{x} = ((t_0, x_0), \ldots, (t_n, x_n)) \in K$. Let $\widehat{X} = \widehat{\iota}(\mathbf{x})$. Now $\|\tau\|_\infty \leq T - \tau$ is bounded so by Lemma B.9 and the assumption of equation (12), there exists a constant $C_1 > 0$ independent of $\mathbf{x}$ such that

$$\left\| \widehat{X} \right\|_\infty + \left\| \frac{\mathrm{d}\widehat{X}}{\mathrm{d}t} \right\|_\infty + \left\| \frac{\mathrm{d}^2 \widehat{X}}{\mathrm{d}t^2} \right\|_\infty < C_1.$$

Thus by Lemma B.8, $\widehat{\iota}(K)$ is relatively compact in $\mathcal{V}^1([\tau, T]; \mathbb{R}^v)$.

Let $K_1 = \overline{\widehat{\iota}(K)}$, where the overline denotes a closure. Now by Theorem B.7, and defining $F^w$ and $\xi^w$ as in the statement of that theorem,

$$\bigcup_{w \in \mathbb{N}} \left\{ \widehat{\iota}(\mathbf{x}) \mapsto \ell(z_T^{f,\zeta,\mathbf{x}}) \;\middle|\; f \in F^w, \ell \in L^{w,u}, \zeta \in \xi^w \right\}$$

is dense in $C(K_1, \mathbb{R}^u)$.

For any $f \in F^w$, any $\zeta \in \xi^w$, any $f_{\mathcal{NN}} \in F_{\mathcal{NN}}^w$ and any $\zeta_{\mathcal{NN}} \in \xi_{\mathcal{NN}}^w$, the terminal values $z_T^{f,\zeta,\mathbf{x}}$ and $z_T^{f_{\mathcal{NN}},\zeta_{\mathcal{NN}},\mathbf{x}}$ may be compared by standard estimates, for example as commonly used in the proof of Picard's theorem. Classical universal approximation results for neural networks [71, 72] then yield that

$$\bigcup_{w \in \mathbb{N}} \left\{ \widehat{\iota}(\mathbf{x}) \mapsto \ell(z_T^{f,\zeta,\mathbf{x}}) \;\middle|\; f \in F_{\mathcal{NN}}^w, \ell \in L^{w,u}, \zeta \in \xi_{\mathcal{NN}}^w \right\}$$

is dense in $C(K_1, \mathbb{R}^u)$.

By the definition of the natural cubic spline topology on $\mathcal{TS}_{[\tau,T]}(\mathbb{R}^v)$, then

$$\bigcup_{w \in \mathbb{N}} \left\{ \mathbf{x} \mapsto \ell(z_T^{f,\zeta,\mathbf{x}}) \;\middle|\; f \in F_{\mathcal{NN}}^w, \ell \in L^{w,u}, \zeta \in \xi_{\mathcal{NN}}^w \right\}$$

is dense in $C(K, \mathbb{R}^u)$. $\qquad\square$

## C   Comparison to alternative ODE models

Suppose if instead of equation (4), we replace $g_{\theta,X}(z,s)$ by $h_\theta(z, X_s)$ for some other vector field $h_\theta$. This might seem more natural. Instead of having $g_{\theta,X}$ be linear in $\mathrm{d}X/\mathrm{d}s$, we take a $h_\theta$ that is potentially nonlinear in the control $X_s$.

Have we gained anything by doing so? It turns out no, and in fact we have lost something. The Neural CDE setup directly subsumes anything depending directly on $X$.

**Theorem C.1.** *Let $\tau, T \in \mathbb{R}$ with $\tau < T$, let $v, w \in \mathbb{N}$ with $v + 1 < w$. Let*

$$
\begin{aligned}
F &= \left\{ f \colon \mathbb{R}^w \to \mathbb{R}^{w \times (v+1)} \,\middle|\, f \text{ is continuous} \right\}, \\
H &= \left\{ h \colon \mathbb{R}^{w-v-1} \times \mathbb{R}^{v+1} \to \mathbb{R}^{w-v-1} \,\middle|\, h \text{ is continuous} \right\}, \\
\xi &= \left\{ \zeta \colon \mathbb{R}^{v+1} \to \mathbb{R}^w \,\middle|\, \zeta \text{ is continuous} \right\}, \\
\mathbb{X} &= \left\{ \widehat{X} \colon [\tau, T] \to \mathbb{R}^v \,\middle|\, \widehat{X} \text{ continuous and of bounded variation} \right\}.
\end{aligned}
$$

*For any $\widehat{X} \in \mathbb{X}$, let $X_t = (\widehat{X}_t, t)$. Let $\pi \colon \mathbb{R}^w \to \mathbb{R}^{w-v-1}$ be the orthogonal projection onto the first $w - v - 1$ coordinates.*

*For any $f \in F$, any $\zeta \in \xi$, and any $\widehat{X} \in \mathbb{X}$, let $z^{f,\zeta,X} \colon [\tau, T] \to \mathbb{R}^w$ be the unique solution to*

$$
z_t^{f,\zeta,X} = z_\tau^{f,\zeta,X} + \int_\tau^t f(z_s^{f,\zeta,X}) \mathrm{d}X_s \quad \text{for } t \in (\tau, T],
$$

*with $z_\tau^{f,\zeta,X} = \zeta(X_\tau)$.*

*Similarly for any $h \in H$, any $\zeta \in \xi$, and any $\widehat{X} \in \mathbb{X}$, let $y^{f,X} \colon [\tau, T] \to \mathbb{R}^{w-v-1}$ be the unique solution to*

$$
y_t^{h,\zeta,X} = y_\tau^{h,\zeta,X} + \int_\tau^t h(y_s^{h,\zeta,X}, X_s) \mathrm{d}s \quad \text{for } t \in (\tau, T],
$$

*with $y_\tau^{h,\zeta,X} = \pi(\zeta(X_\tau))$.*

*Let $\mathcal{Y} = \left\{ \widehat{X} \mapsto y^{h,\zeta,X} \,\middle|\, h \in H, \zeta \in \xi \right\}$ and $\mathcal{Z} = \left\{ \widehat{X} \mapsto \pi \circ z^{f,\zeta,X} \,\middle|\, f \in F, \zeta \in \xi \right\}$.*

*Then $\mathcal{Y} \subsetneq \mathcal{Z}$.*

In the above statement, then a practical choice of $f \in F$ or $h \in H$ will typically correspond to some trained neural network.

Note the inclusion of time via the augmentation $\widehat{X} \mapsto X$. Without it, then the reparameterisation invariance property of CDEs [18], [23, Proposition A.7] will restrict the possible functions that CDEs can represent. This hypothesis is not necessary for the $\mathcal{Y} \neq \mathcal{Z}$ part of the conclusion.

Note also how the CDE uses a larger state space of $w$, compared to $w-v-1$ for the alternative ODE. The reason for this is that whilst $f$ has no explicit nonlinear dependence on $X$, we may construct it to have such a dependence implicitly, by recording $X$ into $v+1$ of its $w$ hidden channels, whereupon $X$ is hidden state and may be treated nonlinearly. This hypothesis is also not necessary to demonstrate the $\mathcal{Y} \neq \mathcal{Z}$ part of the conclusion.

This theorem is essentially an algebraic statement, and is thus not making any analytic claims, for example on universal approximation.

*Proof.*

**That $\mathcal{Y} \neq \mathcal{Z}$:**   Let $z^{f,\zeta,\cdot} \in \mathcal{Z}$ for $\zeta \in \xi$ arbitrary and $f \in F$ constant and such that

$$
f(z) = \left[\begin{array}{ccccc}
1 & 0 & 0 & \cdots & 0 \\
0 & 0 & 0 & \cdots & 0 \\
\vdots & \vdots & \vdots & & \vdots \\
0 & 0 & 0 & \cdots & 0
\end{array}\right]\left.\vphantom{\begin{array}{c}1\\0\\\vdots\\0\end{array}}\right\} w
$$

$$
\underbrace{\phantom{1\ 0\ 0\ \cdots\ 0}}_{v+1}
$$

Then for any $\widehat{X} \in \mathbb{X}$, the corresponding CDE solution in $\mathcal{Z}$ is

$$
z_t^{f,\zeta,X} = z_\tau^{f,\zeta,X} + \int_\tau^t f(z_s^{f,\zeta,X})\mathrm{d}X_s,
$$

and so the first component of its solution is

$$
z_t^{f,\zeta,X,1} = X_t^1 - X_\tau^1 + \zeta^1(X_\tau),
$$

whilst the other components are constant

$$
z_t^{f,\zeta,X,i} = \zeta^i(X_\tau)
$$

for $i \in \{2, \ldots, w\}$, where superscripts refer to components throughout.

Now suppose for contradiction that there exists $y^{h,\zeta,\cdot} \in \mathcal{Y}$ for some $\Xi \in \xi$ and $h \in H$ such that $y^{h,\Xi,X} = \pi \circ z^{f,\zeta,X}$ for all $\widehat{X} \in \mathbb{X}$. Now $y^{h,\Xi,X}$ must satisfy

$$
y_t^{h,\Xi,X} = y_\tau^{h,\Xi,X} + \int_\tau^t h(y_s^{h,\Xi,X}, X_s)\mathrm{d}s,
$$

and so

$$
(X_t^1 - X_\tau^1 + \zeta^1(X_\tau), 0, \ldots, 0) = \pi(\Xi(X_\tau)) + \int_\tau^t h((X_s^1 - X_\tau^1 + \zeta^1(X_\tau), \zeta^2(X_\tau), \ldots, \zeta^w(X_\tau)), X_s)\mathrm{d}s.
$$

Consider those $X$ which are differentiable. Differentiating with respect to $t$ now gives

$$
\frac{\mathrm{d}X^1}{\mathrm{d}t}(t) = h^1((X_s^1 - X_\tau^1 + \zeta^1(X_\tau), \zeta^2(X_\tau), \ldots, \zeta^w(X_\tau)), X_t). \tag{13}
$$

That is, $h^1$ satisfies equation (13) for all differentiable $X$. This is clearly impossible: the right hand side is a function of $t$, $X_t$ and $X_\tau$ only, which is insufficient to determine $\mathrm{d}X^1/\mathrm{d}t(t)$.

**That $\mathcal{Y} \subseteq \mathcal{Z}$:** Let $y^{h,\Xi,X} \in \mathcal{Y}$ for some $\Xi \in \xi$ and $h \in H$. Let $\sigma\colon \mathbb{R}^w \to \mathbb{R}^{v+1}$ be the orthogonal projection onto the last $v+1$ coordinates. Let $\zeta \in \xi$ be such that $\pi \circ \zeta = \pi \circ \Xi$ and $\sigma(\zeta(X_\tau)) = X_\tau$. Then let $f \in F$ be defined by

$$
f(z) = \left[\begin{array}{ccccc}
0 & 0 & \cdots & 0 & h^1(\pi(z), \sigma(z)) \\
\vdots & \vdots & & \vdots & \vdots \\
0 & 0 & \cdots & 0 & h^{w-v-1}(\pi(z), \sigma(z)) \\
1 & 0 & \cdots & 0 & 0 \\
0 & 1 & \cdots & 0 & 0 \\
\vdots & \vdots & \ddots & \vdots & \vdots \\
0 & 0 & \cdots & 1 & 0 \\
0 & 0 & \cdots & 0 & 1
\end{array}\right]
\begin{array}{l}
\left.\vphantom{\begin{array}{c}0\\\vdots\\0\end{array}}\right\} w-v-1 \\
\left.\vphantom{\begin{array}{c}1\\0\\\vdots\\0\\1\end{array}}\right\} v+1
\end{array}
$$

$$
\underbrace{\phantom{0\ 0\ \cdots\ 0}}_{v} \quad \underbrace{\phantom{0}}_{1}
$$

Then for $t \in (\tau, T]$,

$$z_t^{f,\zeta,X} = \zeta(X_\tau) + \int_\tau^t f(z_s^{f,\zeta,X}) \mathrm{d}X_s$$

$$= \zeta(X_\tau) + \int_\tau^t \begin{bmatrix} 0 & 0 & \cdots & 0 & h^1(\pi(z_s^{f,\zeta,X}), \sigma(z_s^{f,\zeta,X})) \\ \vdots & \vdots & & \vdots & \vdots \\ 0 & 0 & \cdots & 0 & h^{w-v-1}(\pi(z_s^{f,\zeta,X}), \sigma(z_s^{f,\zeta,X})) \\ 1 & 0 & \cdots & 0 & 0 \\ 0 & 1 & \cdots & 0 & 0 \\ \vdots & \vdots & \ddots & \vdots & \vdots \\ 0 & 0 & \cdots & 1 & 0 \\ 0 & 0 & \cdots & 0 & 1 \end{bmatrix} \begin{bmatrix} \mathrm{d}\widehat{X}_s^1 \\ \vdots \\ \mathrm{d}\widehat{X}_s^v \\ \mathrm{d}s \end{bmatrix}$$

$$= \zeta(X_\tau) + \int_\tau^t \begin{bmatrix} h^1(\pi(z_s^{f,\zeta,X}), \sigma(z_s^{f,\zeta,X}))\mathrm{d}s \\ \vdots \\ h^{w-v-1}(\pi(z_s^{f,\zeta,X}), \sigma(z_s^{f,\zeta,X}))\mathrm{d}s \\ \mathrm{d}\widehat{X}_s^1 \\ \vdots \\ \mathrm{d}\widehat{X}_s^v \\ \mathrm{d}s \end{bmatrix}$$

$$= \zeta(X_\tau) + \int_\tau^t \begin{bmatrix} h(\pi(z_s^{f,\zeta,X}), \sigma(z_s^{f,\zeta,X}))\mathrm{d}s \\ \mathrm{d}X_s \end{bmatrix}.$$

Thus in particular

$$\sigma(z_t^{f,\zeta,X}) = \sigma(\zeta(X_\tau)) + \int_\tau^t \mathrm{d}X_s = \sigma(\zeta(X_\tau)) - X_\tau + X_t = X_t.$$

Thus

$$\pi(z_t^{f,\zeta,X}) = \pi(\zeta(X_\tau)) + \int_\tau^t h(\pi(z_s^{f,\zeta,X}), \sigma(z_s^{f,\zeta,X}))\mathrm{d}s = \pi(\Xi(X_\tau)) + \int_\tau^t h(\pi(z_s^{f,\zeta,X}), X_s)\mathrm{d}s.$$

Thus we see that $\pi(z^{f,\zeta,X})$ satisfies the same differential equation as $y^{h,\Xi,X}$. So by uniqueness of solution [20, Theorem 1.3], $y^{h,\Xi,X} = \pi(z^{f,\zeta,X}) \in \mathcal{Z}$. □

# D   Experimental details

## D.1   General notes

**Code**   Code to reproduce every experiment can be found at https://github.com/patrick-kidger/NeuralCDE.

**Normalisation**   Every dataset was normalised so that each channel has mean zero and variance one.

**Loss**   Every binary classification problem used binary cross-entropy loss applied to the sigmoid of the output of the model. Every multiclass classification problem used cross-entropy loss applied to the softmax of the output of the model.

**Architectures**   For both the Neural CDE and ODE-RNN, the integrand $f_\theta$ was taken to be a feedforward neural network. A final linear layer was always used to map from the terminal hidden state to the output.

**Activation functions**   For the Neural CDE model we used ReLU activation functions. Following the recommendations of [13], we used tanh activation functions for the ODE-RNN model, who remark that for the ODE-RNN model, tanh activations seem to make the model easier for the ODE

solver to resolve. Interestingly we did not observe this behaviour when trying tanh activations and `method='dopri5'` with the Neural CDE model, hence our choice of ReLU.

**Optimiser**     Every problem used the Adam [73] optimiser as implemented by PyTorch 1.3.1 [74]. Learning rate and batch size varied between experiments, see below. The learning rate was reduced if metrics failed to improve for a certain number of epochs, and training was terminated if metrics failed to improve for a certain (larger) number of epochs. The details of this varied by experiment, see the individual sections. Once training was finished, then the parameters were rolled back to the parameters which produced the best validation accuracy throughout the whole training procedure. The learning rate for the final linear layer (mapping from the hidden state of a model to the output) was typically taken to be much larger than the learning rate used elsewhere in the model; this is a standard trick that we found improved performance for all models.

**Hyperparameter selection**     In brief, hyperparameters were selected to optimise the ODE-RNN baseline, and equivalent hyperparameters used for the other models.

In more detail:

We began by selecting the learning rate. This was selected by starting at 0.001 and reducing it until good performance was achieved for a small ODE-RNN model with batch size 32.

After this, we increased the batch size until the selected model trained at what was in our judgement a reasonable speed. As is standard practice, we increased the learning rate proportionate to the increase in batch size.

Subsequently we selected model hyperparameters (number of hidden channels, width and depth of the vector field network) via a grid search to optimise the ODE-RNN baseline. A single run of each hyperparameter choice was performed. The equivalent hyperparameters were then used on the GRU-$\Delta$t, GRU-D, GRU-ODE baselines, and also our Neural CDE models, after being adjusted to produce roughly the same number of parameters for each model.

The grids searched over and the resulting hyperparameters are stated in the individual sections below.

**Weight regularisation**     $L^2$ weight regularisation was applied to every parameter of the ODE-RNN, GRU-$\Delta$t and GRU-D models, and to every parameter of the vector fields for the Neural CDE and GRU-ODE models.

**ODE Solvers**     The ODE components of the ODE-RNN, GRU-ODE, and Neural CDE models were all computed using the fourth-order Runge-Kutta with 3/8 rule solver, as implemented by passing `method='rk4'` to the `odeint_adjoint` function of the `torchdiffeq` [24] package. The step size was taken to equal the minimum time difference between any two adjacent observations.

**Adjoint backpropagation**     The GRU-ODE, Neural CDE and the ODE component of the ODE-RNN are all trained via the adjoint backpropagation method [15], as implemented by `odeint_adjoint` function of the `torchdiffeq` package.

**Computing infrastructure**     All experiments were run on one of two computers; both used Ubuntu 18.04.4 LTS, were running PyTorch 1.3.1, and used version 0.0.1 of the `torchdiffeq` [24] package. One computer was equipped with a Xeon E5-2960 v4, two GeForce RTX 2080 Ti, and two Quadro GP100, whilst the other was equipped with a Xeon Silver 4104 and three GeForce RTX 2080 Ti.

### D.2   CharacterTrajectories

The learning rate used was 0.001 and the batch size used was 32. If the validation loss stagnated for 10 epochs then the learning rate was divided by 10 and training resumed. If the training loss or training accuracy stagnated for 50 epochs then training was terminated. The maximum number of epochs allowed was 1000.

We combined the train/test split of the original dataset (which are of unusual proportion, being 50%/50%), and then took a 70%/15%/15% train/validation/test split.

The initial condition $\zeta_\theta$ of the Neural CDE model was taken to be a learnt linear map from the first observation to the hidden state vector. (Recall that is an important part of the model, to avoid translation invariance.)

The hyperparameters were optimised (for just the ODE-RNN baseline as previously described) by performing most of a grid search over 16 or 32 hidden channels, 32, 48, 64, 128 hidden layer size, and 1, 2, 3 hidden layers. (The latter two hyperparameters corresponding to the vector fields of the ODE-RNN and Neural CDE models.) A few option combinations were not tested due to the poor performance of similar combinations. (For example every combination with hidden layer size of 128 demonstrated relatively poor performance.) The search was done on just the 30% missing data case, and the same hyperparameters were used for the 50% and 70% missing data cases.

The hyperparameters selected were 32 hidden channels for the Neural CDE and ODE-RNN models, and 47 hidden channels for the GRU-$\Delta$t, GRU-D and GRU-ODE models. The Neural CDE and ODE-RNN models both used a feedforward network for their vector fields, with 3 hidden layers each of width 32. The resulting parameter counts for each model were 8212 for the Neural CDE, 8436 for the ODE-RNN, 8386 for the GRU-D, 8292 for the GRU-$\Delta$t, and 8372 for the GRU-ODE.

### D.3 PhysioNet sepsis prediction

The batch size used was 1024 and learning rate used was 0.0032, arrived at as previously described. If the training loss stagnated for 10 epochs then the learning rate was divided by 10 and training resumed. If the training loss or validation accuracy stagnated for 100 epochs then training was terminated. The maximum number of epochs allowed was 200. The learning rate for the final linear layer (a component of every model, mapping from the final hidden state to the output) used a learning rate that 100 times larger, so 0.32.

The original dataset does not come with an existing split, so we took our own 70%/15%/15% train/validation/test split.

As this problem featured static (not time-varying) features, we incorporated this information by allowing the initial condition of every model to depend on these. This was taken to be a single hidden layer feedforward network with ReLU activation functions and of width 256, which we did not attempt a hyperparameter search over.

As this dataset is partially observed, then for the ODE-RNN, GRU-$\Delta$t, GRU-D models, which require *something* to be passed at each time step, even if a value is missing, then we fill in missing values with natural cubic splines, for ease of comparison with the Neural CDE and ODE-RNN models. (We do not describe this as imputation as for the observational intensity case the observational mask is additionally passed to these models.) In particular this differs slightly from the usual implementation of GRU-D, which usually use a weighted average of the last observation and the mean. Splines accomplishes much the same thing, and help keep things consistent between the various models.

The hyperparameters were optimised (for just the ODE-RNN baseline as previously described) by performing most of a grid search over 64, 128, 256 hidden channels, 64, 128, 256 hidden layer size, and 1, 2, 3, 4 hidden layers. (The latter two hyperparameters corresponding to the vector fields of the ODE-RNN and Neural CDE models.)

The hyperparameters selected for the ODE-RNN model were 128 hidden channels, and a vector field given by a feedforward neural network with hidden layer size 128 and 4 hidden layers. In order to keep the number of parameters the same between each model, this was reduced to 49 hidden channels and hidden layer size 49 for the Neural CDE model, and increased to 187 hidden channels for the GRU-$\Delta$t, GRU-D and GRU-ODE models. When using observational intensity, the resulting parameter counts were 193541 for the Neural CDE, 194049 for the ODE-RNN, 195407 for the GRU-D, 195033 for the GRU-$\Delta$t, and 194541 for the GRU-ODE. When not using observational intensity, the resulting parameter counts were 109729 for the Neural CDE, 180097 for the ODE-RNN, 175260 for the GRU-D, 174886 for the GRU-$\Delta$t, and 174921 for the GRU-ODE. Note the dramatically reduced parameter count for the Neural CDE; this is because removing observational intensity reduces the number of channels, which affects the parameter count dramatically as discussed in Section 6.3.

### D.4 Speech Commands

The batch size used was 1024 and the learning rate used was 0.0016, arrived at as previously described. If the training loss stagnated for 10 epochs then the learning rate was divided by 10 and

training resumed. If the training loss or validation accuracy stagnated for 100 epochs then training was terminated. The maximum number of epochs allowed was 200. The learning rate for the final linear layer (a component of every model, mapping from the final hidden state to the output) used a learning rate that 100 times larger, so 0.16.

Each time series from the dataset is univariate and of length 16000. We computed 20 Mel-frequency cepstral coefficients of the input as implemented by `torchaudio.transforms.MFCC`, with logarithmic scaling applied to the coefficients. The window for the short-time Fourier transform component was a Hann window of length 200, with hop length of 100, with 200 frequency bins. This was passed through 128 mel filterbanks and 20 mel coefficients extracted. This produced a time series of length 161 with 20 channels. We took a 70%/15%/15% train/validation/test split.

The hyperparameters were optimised (for just the ODE-RNN baseline as previously described) by performing most of a grid search over 32, 64, 128 hidden channels, 32, 64, 128 hidden layer size, and 1, 2, 3, 4 hidden layers. (The latter two hyperparameters corresponding to the vector fields of the ODE-RNN and Neural CDE models.)

The hyperparameters selected for the ODE-RNN model were 128 hidden channels, and a vector field given by a feedforward neural network with hidden layer size 64 and 4 hidden layers. In order to keep the number of parameters the same between each model, this was reduced to 90 hidden channels and hidden layer size 40 for the Neural CDE model, and increased to 160 hidden channels for the GRU-$\Delta$t, GRU-D and GRU-ODE models. The resulting parameter counts were 88940 for the Neural CDE model, 87946 for the ODE-RNN model, 89290 for the GRU-D model, 88970 for the GRU-dt model, and 89180 for the GRU-ODE model.