[Reviews · NeurIPS 2020]

Review 1

Summary and Contributions: This paper addresses the problem of incorporating input observations into the Neural ODE framework. It introduces the theory of controlled differential equations (CDE), which allow dependence on arbitrary input functions in a continuous manner, in contrast to prior work on integrating Neural ODEs with discrete inputs. The method can process incoming data that may be irregularly sampled and partially observed and can be trained with a single instance of adjoint-based backprop across observations. Experiments were performed on a number of time series datasets including trajectory classification and speech recognition.

Strengths: - The proposed model addresses a clear problem and proposes a clear solution. The solution is simple and motivated and theoretically grounded. - The background and method are explained well, as well as comparisons to prior work. - The method is novel and relevant to the NeurIPS community

Weaknesses: - All experiments were somewhat toy, but this is understandable given the nature of the contributions

Correctness: The method and the empirical methodology are sound

Clarity: The paper is very well written and easy to follow

Relation to Prior Work: Prior work is clearly discussed and compared against

Reproducibility: Yes

Additional Feedback: After author response and discussion: ============================ After reading the author response and discussing with the other reviewers, I have the following additional points to add: (*) I agree with R3's concerns about the spline preprocessing, which seems to be a significant point of departure from prior related methods that wasn't explained and ablated sufficiently. (*) I'm concerned with some of the baselines. As I understand, the GRU-D and GRU-Delta baselines should *not* be given access to the interpolated data, but from looking at the code it seems to me that they are (*) To isolate the effect of the spline itself, I think the experiment that R3 suggested of having a *pure RNN* baseline (e.g. GRU instead of GRU-D) that operates on the *splined* data is important (*) The Informal Theorem (L117) leaves out another natural comparison: why not compare against $h_\theta(z_s, X'_s)$ instead of $h_\theta(z_s, X_s)$? I like the explanation that the latter has weaker representation power, but I think a discussion (and ablation) of the former is quite natural to understand where the benefits of Neural CDE comes from. Given that you're already adding an additional spline step which gives access to X'_s, the former (i) seems to be a more natural extension of neural ODE, (ii) is more motivated and doesn't require the CDE formalism, and (iii) has stronger representation power than neural CDE. I apologize for not raising these concerns in my initial review. As such I am keeping my score, but I hope that this feedback helps the authors refine the paper.


Review 2

Summary and Contributions: This paper incorporates controlled differential equations into the NeuralODE model and proposes a new model named Neural Controlled Differential Equation (NCDE) which can operate directly on irregularly sampled and partially observed multivariate time series. NCDE can be trained in a memory efficient way with the adjoint method. The authors demonstrate that NCDE outperforms state-of-the-art ODE/RNN based models on the CharacterTrajectories, PhysioNet, and Speech Commands datasets. Overall, this paper could be a significant algorithmic contribution, with the caveat for some clarifications. Given these clarifications in an author response, I would be willing to increase the score.

Strengths: The idea of incorporating controlled differential equations into the NeuralODE model is novel and interesting. The authors have clearly explained the method, as well as the background works. Empirical results from the paper are convincing, and the experiments are properly designed.

Weaknesses: There are some unclear parts in the paper. First, how is the NCDE trained? What are the training losses? Second, what does the model look like? Does it just contain a feedforward step as in RNNs or it also consists of an additional reconstruction step as in VAEs? In addition, it would be more convincing if the paper provides an analysis on the number of function evaluations required by the solvers in the NCDE compared to those in other ODE-based methods such as the GRU-ODE.

Correctness: The claims and method are correct. The empirical methodology is correct.

Clarity: The paper is well-written.

Relation to Prior Work: It is clearly discussed how the proposed method differs from previous works.

Reproducibility: Yes

Additional Feedback: Below are additional questions that I have: - What is the effect of error tolerance of the ODE solvers in the forward and backpropagation step on the performance of the NCDE? - How stable is the training of the NCDE? Post Rebuttal Comments: The rebuttal addressed my concerns, and I am happy with the clarifications and additional results on the number of function evaluations of the proposed methods and other baselines. I do agree with Reviewer 3 that a baseline involving cubic splines alone is an important ablation. After carefully reading the rebuttal, the reviews, and the discussion, I decided not to change my score (6) for this paper.


Review 3

Summary and Contributions: The paper proposes a new framework for irregular time series data. The authors combine Neural ODEs with cubic splines, allowing to avoid some of the issues of the previous ODE models. =============================== Added after author response: thank you for providing an example of oscillatory data modelled by cubic splines and results on other interpolation schemes. The authors addressed all my questions. I think the paper has a strong contribution and solves important issues of the original paper. I have increased a score to 7:accept. I am still a bit skeptical about the fact that splines are doing the heavy-lifting of constructing a smooth version of the trajectory instead of Neural ODEs, although the authors verified experimentally that results are consistent across interpolation schemes. It would be interesting to see the case when the splines do not match the original trajectory from which the observations were sampled, perhaps, in case of very noisy data. I am looking forward to the follow-up work (mentioned in the author response) and more investigation on the role of interpolation schemes.

Strengths: Adding splines to Neural ODEs is an interesting idea. It allows to query the function at any time point without the need to stop and update the latent state at every observation. It also allows to overcome batching issues and reduce memory consumption. This makes the ODE-based models easier to fit and more usable in practice.

Weaknesses: The cubic spline introduce the inductive bias about the smoothness of the input, which might or might not be appropriate for different applications. In CharacterTrajectories and Physionet applications, the data is fairly smooth, and I am wondering if the cubic splines alone (without Neural ODE) might be good enough to perform forecasting and classification. In the Speech Commands experiment, I am assuming that the input trajectories are not smooth, and applying cubic splines will misrepresent the data. But, according to the experiments in the paper, Neural CDE still perform well on this dataset. Do authors have any explanation for this phenomenon? How do the cubic splines fitted to the Speech Commands data look like? Finally, the model might be sensitive to the choice of the spline. Have authors experimented with other types of splines?

Correctness: I am concerned about the decision to add the index of the observation to the input for Neural CDE. It is unfair to explicitly give the number of the observations to the Neural CDE, but not to other models (ODE-RNN, GRU-D, and GRU-t) (lines 209-213). In Physionet dataset the number of observations in the time series may be correlated with the sepsis prediction (if patient goes into sepsis state, more measurements were taken). Can it be the reason for such a drastic performance difference with and without OI for Neural CDE model (0.88 versus 0.776, Table 2). The same type of informations needs to be added to the input of other models as well.

Clarity: The paper is clearly written and easy to follow.

Relation to Prior Work: The relation to the prior work is adequately described.

Reproducibility: Yes

Additional Feedback: The authors have compared the ODE-based methods only on time series classification tasks, which is surprising. From the results in Rubanova et al. (2019), ODE-based models are beneficial in the forecasting and interpolation tasks, but performed similarly to RNN-based models in classification task on Physionet. Was there a reason behind using only classification tasks for this paper? I am guessing that Neural CDEs might have an advantage over other ODE models in forecasting and interpolation tasks thanks to the cubic splines.


Review 4

Summary and Contributions: "Neural Controlled Differential Equations for Irregular Time Series" describes a method for adapting the recently proposed Neural ODE to time series modeling, specifically continuous time as in the underlying generation processes (pre-discretization or binning). Extending this approach to controlled differential equations, the authors introduce an extention dubbed neural controlled differential equations (neural CDE).

Strengths: The background, derivation, and core concept of the neural CDE are clear, and novel to my knowledge, building on a strong foundation of recent work in neural ODEs while bridging the gap to a wealth of other methods in differential equations. Building on this, the core theoretical definitions and benefits are described without excess jargon, and should be accessible to most readers familiar with time-series modeling and the basics of neural ODE. The experimental section shows numerous examples of the benefits on neural CDEs for time-series modeling, and directly highlights the reasons when and why neural CDEs might be beneficial in practice. I also commend the authors on the (to be released) code - this will help many others follow up on applying this promising new method in other areas, and applying to experiments such as the above mentioned. The appendix is also extensive, and useful for further understanding the content of the core paper.

Weaknesses: Reduction of the formal proofs of section 3 in the main paper (relegated to the appendix, which is quite thorough) is unfortunate, but seems necessary for space considerations. It is worth considering a more "smooth" way to perform these reductions, as the subsection headings and definition of informal theorems, followed by a 1 line description feels odd to me. The descriptions are very useful, but there may be a way to do the same but continue the "flow" of the paper better - I leave this to the authors' discretion. Experiments compare to several other relevant and alternative methods, and clearly discuss and show the benefits of the introduced method. The main critique here is that the datasets in question are somewhat esoteric, though I do not know which "more common" datasets might be relevant while still highlighting the special abilities of neural CDE, given the specific type of data necessary and design of the model itself. As simple as it sounds, showing a picture of the CharacterTrajectories data setup could help readers less familiar with the general views of "time-series" style handwriting data. Application to speaker recognition, as opposed to speech commands would demonstrate usage of the model across another "axis/dimension" of the speech problem - this is not necessary but tying to a more common dataset could help motivate additional readers. It may also be easier to motivate "irregular sampling" for speaker recognition using multi-scale recordings, "lost segments" of a larger sentence, only having low-quality recordings for one speaker vs high quality for another, or other "deconstructions" of base data. This is *not at all* necessary for potential publication of this paper, but further experiments are always worth considering if the authors have any further ideas (and the time/compute to execute, of course).

Correctness: The method is well motivated, and the experimental methodology clearly shows the high level concept and key benefits of the approach.

Clarity: This paper is clearly written, strongly motivated, excellently framed, innovates a new model, demonstrates both its theoretical generalization and application, all while discussing practical concerns and releasing code for reproduction, extension, and investigation. I commend the authors on a very nice paper, which was a pleasure to read.

Relation to Prior Work: The background and motivation for the paper are clear, referencing and grounding the upcoming discussions in relevant related work, methodology from differential equations, and even relevant textbooks. Mathematical tools and techniques are introduced succinctly, and the overall "stage setting" of the paper is well done.

Reproducibility: Yes

Additional Feedback: After feedback: The feedback to the other reviewer's questions was very useful, and the proposed changes should further strengthen the paper.

[Author Response · NeurIPS 2020]

We thank all our reviewers for their time and for their kind words. We are delighted to receive such uniformly positive reviews.

**Reviewer 1** comments that the experiments could be on larger problems – we agree that this is fair, and this is certainly of interest going forward.

**Reviewer 2** comments that some details are unclear. Apologies, to clarify, the loss function was binary cross entropy for the PhysioNet dataset, and the usual cross entropy for CharacterTrajectories and Speech Commands. Training was via SGD in the usual manner. These details were put in the appendix for space; we will add them to the main paper. As you suggest, the model implements a feedforward step analogous to an RNN. An additional reconstruction step analogous to a VAE would be possible if desired, but wasn't something we explored here.

On the number of function evaluations (NFE), leaving this out was an oversight. Running the experiments with `method='dopri5', rtol=1e-4, atol=1e-6` we observe:

|  | Speech Commands | Sepsis (OI) | Sepsis (No OI) | CharacterTrajectories 30% |
|---|---|---|---|---|
| Neural CDE | **6009 ± 494** | 2872 ± 158 | 3123 ± 223 | **1955 ± 233** |
| GRU-ODE | 8579 ± 1355 | 3175 ± 194 | 3304 ± 178 | 1974 ± 110 |
| ODE-RNN | 6399 ± 2 | **2472 ± 127** | **2681 ± 47** | 7242 ± 6 |

CharacterTrajectories 50%, 70% are similar to 30% and so are not shown. The high mean and low variance of the NFEs for ODE-RNN on some problems is likely due to the high sampling rate of the data.

Regarding error tolerances, we have observed successful training, in the sense of high classification accuracies, across a range of tolerances ranging from $10^{-3}$ down to $10^{-8}$, even for oscillatory problems such as the Speech Commands dataset. Generally speaking Neural CDEs seem to be slightly harder (but not dramatically harder) to solve than a comparable Neural ODE, no doubt due to the time-varying nature of the problem.

Regarding stability of training, Neural CDEs seem to be unusually stable. We found that they always seem to train well, regardless of the choice of optimiser and learning rate, even when competing methods (GRUs etc.) are prone to fail. We make a brief remark about this in the paper, but frankly this isn't a phenomenen we understand yet.

**Reviewer 3** asks about the smoothness prior of splines. Splines can actually represent very oscillatory data without issues. See the figure to the right, which is a cubic spline through one channel of a sample from the Speech Commands dataset (and it is a typical such sample). Integrating this simply requires putting down enough points, and moreover this is not computationally daunting.

We have indeed experimented with other interpolation schemes. We find that performance differences are minor; trained models have similar classification accuracies. We do however observe that linear interpolation produces a model that requires fewer function evalutions on the forward pass, but increased function evaluations on the backward pass. Furthermore linear interpolation is causal whilst cubic splines are non-causal. This is a topic we are performing follow-up work on presently.

Regarding the correctness of the experiments, the other models received the same information (in particular the observational intensity) that the Neural CDE did – we appreciate that to have done otherwise would have been unfair. The paper does word this a little oddly (essentially we sought to emphasise the difference between $X$ and $\mathrm{d}X/\mathrm{d}t$), so we will ensure that it is clear on this point.

Regarding the choice of only considering classification tasks – there was no real reason for this choice, and we expect Neural CDEs to be able to perform regression, forecasting, etc. as well. We completely agree it would be valuable to consider other problems; were we to do this again (with the benefit of hindsight) then we probably would. Each experiment was chosen to demonstrate a particular type of problem: CharacterTrajectories for varying irregular data; PhysioNet for observational intensity and partially observed data; Speech Commands for regular data.

**Reviewer 4** gives a truly delightful review – thank you! We are very happy that our paper is identified as making both significant theoretical and practical contributions. Regarding the brevity of the proof descriptions, we will aim to use part of the additional page to give more detail of these.

On the choice of datasets, these were intended to be standard choices. CharacterTrajectories is drawn from the UEA archive, PhysioNet is a central resource for medical time series, and Speech Commands is now provided through `torchaudio`. We agree that a sample from the datasets could be informative to fix what is going on in a reader's mind; if space allows we will add this to the paper. Thankyou for the suggestions of other speech problems – we're always looking for good ideas for follow up work.

[Meta-Review · NeurIPS 2020]

The paper extends Neural ODE for considering input observations and incorporating them in a continuous manner. Inputs can then be irregularly sampled and partially observed. Experiments are performed for classification tasks on a number of time series datasets. Continuous functions are built by using interpolation schemes – here splines. All the reviewers agree that this is a strong contribution extending the use of Neural ODE idea to additional settings and possibly overcoming some issues of the original approach, e.g. computational efficiency. The paper is well motivated and the approach theoretically grounded. The reviewers agree that the authors clarified several issues in their response. The authors are encouraged to use the additional page to incorporate the additional information discussed in the response, e.g. regarding the technical proof (section 3, R4) and the role of splines (e.g the baseline spline + RNN as suggested by R3, R1).